# Acute kidney injury in patients with Visceral Leishmaniasis in Northwest Ethiopia

**Workagegnehu Hailu** [1]☯*, **Rezika Mohamed**[1,2]☯, **Helina Fikre**[2], **Saba Atnafu**[2], **Azeb Tadesse**[2], **Ermias Diro**[1,2]‡, **Johan van Grienvsen**[3]‡

**1** Department of Internal Medicine, College of Medicine and Health Science, University of Gondar, Gondar, Ethiopia, **2** Leishmaniasis Research and Treatment Center, College of Medicine and Health Science, University of Gondar, Gondar, Ethiopia, **3** Department of Clinical Sciences, Institute of Tropical Medicine, Antwerp, Belgium

☯ These authors contributed equally to this work.
‡ These authors also contributed equally to this work.
* workhailu@yahoo.com

## Abstract

### Background

Visceral Leishmaniasis (VL) is a neglected tropical disease endemic to several countries including Ethiopia. Outside of Africa, kidney involvement in VL is frequent and associated with increased mortality. There is however limited data on acute kidney injury (AKI) in VL patients in East-Africa, particularly in areas with high rates of HIV co-infection. This study aims to determine the prevalence, characteristics and associated factors of AKI in VL patients in Northwest Ethiopia.

### Methods

A hospital based retrospective patient record analysis was conducted including patients treated for VL from January 2019 to December 2019 at the Leishmaniasis Research and Treatment Center (LRTC), Gondar, Ethiopia. Patients that were enrolled in ongoing clinical trials at the study site and those with significant incomplete data were excluded. Data was analyzed using SPSS version 20. *P* values were considered significant if < 0.05.

### Results

Among 352 VL patients treated at LRTC during the study period, 298 were included in the study. All were male patients except two; the median age was 23 years (IQR: 20–27). The overall prevalence of AKI among VL patients was 17.4% (confidence interval (CI): 13.6%-22.2%). Pre-renal azotemia (57%) and drug-induced AKI (50%) were the main etiologies of AKI at admission and post-admission respectively. Proteinuria and hematuria occurred in 85% and 42% of AKI patients respectively. Multivariate logistic regression revealed HIV co-infection (adjusted odds ratio (AOR): 6.01 95% CI: 1.99–18.27, p = 0.001) and other concomitant infections (AOR: 3.44 95% CI: 1.37–8.65, p = 0.009) to be independently associated with AKI.

**Data Availability Statement:** The dataset for "Acute Kidney Injury in patients with Visceral Leishmaniasis in Northwest Ethiopia" is published in DRYAD. (doi:10.5061/dryad.v41ns1rw7).

**Funding:** The authors received no specific funding for this work.

**Competing interests:** The authors have declared that no competing interests exist.

## Conclusion

AKI is a frequent complication in Ethiopian VL patients. Other renal manifestations included proteinuria, hematuria, and pyuria. HIV co-infection and other concomitant infections were significantly associated with AKI. Further studies are needed to quantify proteinuria and evaluate the influence of AKI on the treatment course, morbidity and mortality in VL patients.

## Introduction

Leishmaniasis is a neglected parasitic tropical disease caused by various species of *Leishmania*. The diseases is endemic in more than 88 countries, with 350 million people at risk of infection [1]. Visceral Leishmaniasis (VL) is the most serious form of the disease with more than 95% of mortality if left untreated [2]. According to the 2017 WHO report, 94% of new cases of VL cases occurred in seven countries: Brazil, Ethiopia, India, Kenya, Somalia, South Sudan and Sudan [3]. The disease is endemic in numerous foci of the East African countries including Ethiopia, Kenya, South Sudan and Uganda [4]. VL in Ethiopia puts over 3.2 million people at risk, with up to 3000 new cases per year [1, 5]. The number of cases is associated with seasonal migration of labor force to endemic areas and HIV/AIDS [6].

Outside of East-Africa, renal involvement in VL has been found to be frequent and to be associated with increased mortality [7, 8]. Clinical features of renal involvement in VL are diverse including urinary abnormalities (proteinuria, hematuria, and pyuria), tubular dysfunction, glomerular dysfunction, and acute kidney injury (AKI) [9, 10]. AKI is a clinical syndrome characterized by a sudden decrease in glomerular filtration rate (GFR) sufficient to decrease the elimination of nitrogenous waste products and other uremic toxins [11]. Possible causes of development of AKI in VL patients are multifactorial which include the disease itself, drug toxicity, presence of associated infections and hemodynamic abnormalities [8, 12–16].

Different histological patterns of renal disease were reported in patients with VL including interstitial nephritis, proliferative glomerulonephritis, segmental necrotizing glomerulonephritis, and AA amyloid glomerular deposits [17–21].

Studies on renal abnormalities in East-Africa are very scarce. In a study done to assess the efficacy and safety of meglumine antimoniate, 9/54(30%) of Ethiopian VL patients developed abnormal renal function (defined as serum creatinine (SCr) >1.1mg/dl) [22]. Several studies on AKI have been done outside Africa where HIV prevalence is low and Sodium Stibogluconate (SSG), a potentially nephrotoxic drug, is the first line therapy. In Ethiopia, where SSG and Paromomycin (PM) combination is used as a first line treatment and where there is a high rate of HIV co-infection, there is scarcity of data on the actual prevalence of AKI in VL patients. The aim of this study was to determine the prevalence of AKI, characterize its manifestations and associated factors among VL patients in Northwest Ethiopia.

## Methods

### Study design and setting

This is a retrospective analysis of patients' clinical data of VL patients treated at the Leishmaniasis Research and Treatment Center (LRTC) of the University of Gondar Hospital (UoGH) between January 2019 and December 2019. Data were retrieved from medical records over a period of two weeks, from February 3–15, 2020. UoGH is a tertiary-level teaching specialized public hospital located in Gondar town, Northwest Ethiopia. The hospital serves as the only

referral hospital for patients coming from Northwest part of the country serving a catchment area for about 7 million populations. LRTC is located within UoGH and supported by the Drugs for Neglected Diseases Initiative (DNDi). LRTC has 24 inpatients beds and well equipped laboratory and pharmacy services. LRTC is one of the few clinical trial sites in Ethiopia and gives free diagnostic, treatment and follow-up service for VL patients presenting to the hospital. Diagnostic tests for VL including splenic and bone marrow aspiration and culture are routinely performed.

Close monitoring of patient's response and safety are routine practice which includes monitoring of renal function tests, liver function tests and complete blood count (CBC) on weekly basis and when indicated. All patient clinical records including history, physical findings, laboratory tests, type of treatment and final outcome are documented on the medical charts of patients and kept in the center. The first line treatment for VL is SSG + PM and liposomal amphotericin B (L-AMB) is given for patients with critical illness, severe organ dysfunction and HIV co- infection.

## Population, sample size, and sampling procedure

Medical charts of all patients admitted during the study period at LRTC with a diagnosis of VL were retrieved. Diagnosis of VL was made by either the presence of amastigotes on splenic or bone marrow aspirate—Giemsa-stain under light microscopy or a patient fulfilling WHO case definition for VL and positive rk-39 test if tissue aspiration was not done. Exclusion criteria were patients participating in ongoing clinical trials, if serum creatinine was not measured and if patient files could not be retrieved at LRTC chart room [23].

Sample size was calculated with an assumption of 50% of VL patients to have AKI at presentation or during treatment with 90% CI and 5% marginal error, yielding a sample size of 270. Accounting for 10% incomplete data, the total sample size was 297. Patients were included by retrograde consecutive sampling from December 2019 backward until the required sample size was achieved. Patients treated during this period were selected because this provides more recent data, representing the current situation in the study setting.

## Variables and operational definitions

The outcome variable of interest was presence of AKI. The independent variables were age and sex, duration of illness, concomitant illness, cytopenia's, HIV status, and treatment outcome of VL.

AKI definition and staging was as per the Kidney Disease Initiative Global Outcome (KDIGO) guideline [24]. AKI was defined as an increase in serum creatinine (SCr) by ≥0.3 mg/dl within 48 hours after admission; or an increase in SCr to ≥1.5 times baseline, which is known or presumed to have occurred within the prior 7 days. AKI was graded as stage 1 (SCr 1.5–1.9 times baseline or ≥0.3 mg/dl increase), stage 2 (SCr 2.0–2.9 times baseline) and stage 3 (SCr 3.0 times baseline or SCr to ≥4.0 mg/dl).

Resolved AKI was defined as normalization of SCr at discharge (to ≤1.1mg/dl) and non-resolved AKI was defined as not complete normalization of SCr. The AKI status was labelled as unknown if the SCr level was not known at the time of VL outcome assessment. Renal outcome was defined as good if the AKI resolved at discharge and bad if AKI had not resolved, was unknown or death in a patient with AKI not resolved at the time of death. The likely cause of AKI was based on clinical judgment of the treating physician.

VL clinical cure was defined as a patient showing improvement in signs and symptoms after standard treatment (fever resolution, hemoglobin increase, weight gain, and spleen size regression). VL parasitological cure was defined as improvement in signs and symptoms after

standard treatment (fever resolution, hemoglobin increase, weight gain, and spleen size regression) and the absence of parasites in tissue aspirates. Treatment failure was defined as a positive test of cure (parasitological failure) and/or clinical signs/symptoms that persisted after completion standard treatment. Poor VL treatment outcomes included death, treatment failure or unknown outcome and good VL treatment outcome was defined as clinical or parasitological cure.

### Data collection tool and analysis

Data were entered into Microsoft XL version 2016 and were exported to SPSS version 20 for statistical analysis. Data cleaning was performed before conducting descriptive analysis. Baseline characteristics were described using number and percentages. To examine the difference between independent variables among patients with or without AKI binary regression was performed. Both bivariate and multivariate logistic regression analyses were used to identify independently associated factors of AKI in VL patients. Those variables with a P-value <0.2 in the bivariate analysis were exported to multivariate analysis to control the possible effect of confounders. The Chi$^2$ and Fisher exact test were used for binary/categorical variables and the Kruskal Wallis test was used for continuous variables. *P* values were considered significant if < 0.05. Missing data were not included in the analysis.

### Ethical consideration

Ethical clearance and waiver of individual informed consent for this retrospective data analysis was obtained from the University of Gondar Institutional Review Board. Patient data confidentiality was respected at all levels from record review, chart retrieving and data analysis which was handled by the investigators.

## Results

During the period between January 2019 and December 2019, a total of 1603 patients were screened and 352 were treated for VL at LRTC. Of these, 33 were included in ongoing clinical trials, two were excluded because of incomplete data and 19 patient files could not be retrieved. Finally, a total of 298 patients were included in the study for analysis (**Fig 1**)

### Baseline demographic and clinical characteristics

All patients were male except two. The median age was 23 (IQR: 20–27) years. Almost all (97%) of the patients were in the age group between 15–45 years. The median duration of illness was one (IQR: 0.7–2) month. Patients were either referred from nearby health facilities or presented to UoG hospital by themselves. There were no patients who had received anti-leishmanial drugs before admission to the hospital.

More than half of the patients (53%) presented within one to three months after symptom onset. Around 70% of the patients had concomitant diagnoses like tuberculosis, intestinal parasitosis, pneumonia, otitis media and others. A total of 17 patients (6.1%) had VL-HIV co-infection, of which half of the cases (53%; 9/17) were on antiretroviral treatment at the time of VL diagnosis. The majority (95%) of patients had a first episode (primary) VL. Most patients (77.9%) were treated with a combination of SSG and PM for 17 days (**Table 1**).

### Baseline laboratory characteristics

More than 95% of patients had anemia, leukopenia or thrombocytopenia. The majority (94.5%) of patients had low serum albumin; elevated transaminases was also a common

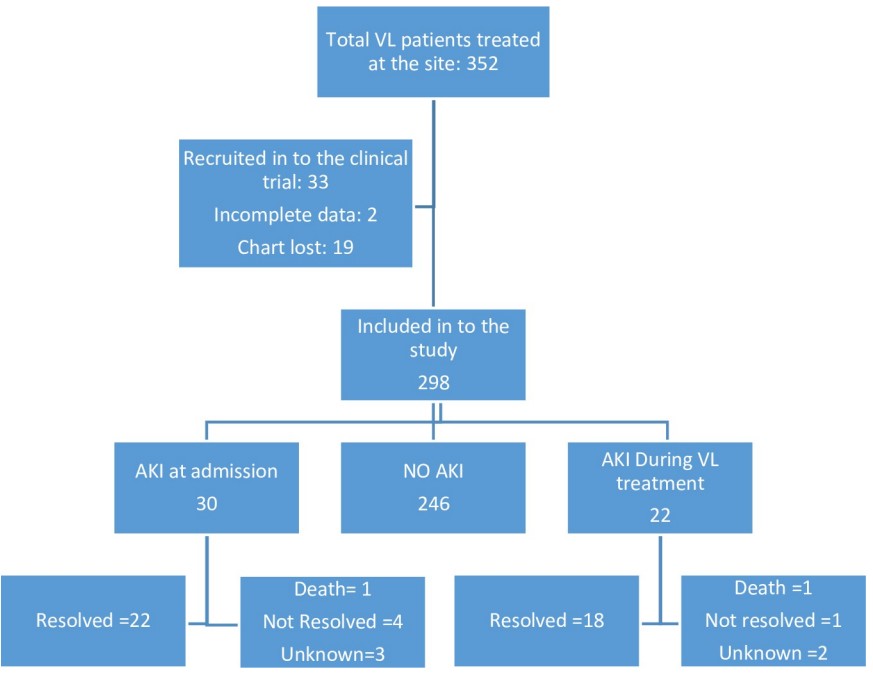

**Fig 1. Patient flow chart.**

finding. Renal manifestations like proteinuria, hematuria and pyuria were seen in 56.9%, 30.3% and 40.9% of VL patients respectively (**Table 2**).

## Acute kidney injury in VL patients

Out of the 298 participants, a total of 52 (17.4%, confidence interval (CI): 13.6% - 22.2%) patients developed AKI. Of these, 30/298 (10%) had AKI at admission before treatment initiation and 22/268 (8.2%) patients developed AKI during the VL treatment course (11 during the first week, eight during the second week and two during the third week of treatment). Two third (66%) of patients had stage 1 AKI. Proteinuria occurred in 58% of patients with AKI while hematuria and pyuria were seen in 42% and 51% respectively (**Table 3**).

## Factors associated with AKI

All baseline characteristics of VL patients were similar for patients with or without AKI except for the presence of HIV infection and concomitant infections, which tended to occur more commonly in patients with AKI. 85% of patients with AKI had concomitant infection while 67% of patients without AKI had concomitant infections (p = 0.0104). HIV co-infection was found in 17% and 4% of patients with and without AKI respectively (p = 0.0025) (**Table 1**). On bivariate analysis, factors significantly associated with AKI were duration of illness, presence of HIV co-infection and concomitant infections. On multivariate analysis, presence of HIV co-infection (adjusted odds ratio (AOR) = 6.01 95% CI: 1.99–18.27, p = 0.001) and other concomitant infections (AOR = 3.44 95% CI: 1.37–8.65, p = 0.009) were found to be associated with the development of AKI.

## Drug induced AKI (post-admission AKI)

A total of 22 patients developed AKI after admission, yielding an incidence of 8.2% (22/268). The proportion of AKI per treatment allocation is shown in **Table 4**.

**Table 1. Baseline demographic and clinical characteristics of VL patients at LRTC between January 2019 –December 2019, Gondar, Ethiopia, N = 298.**

| Variables | Total N (%) | AKI | No AKI | P value |
|---|---|---|---|---|
| | | N (%) | N (%) | |
| **Age, median (IQR); (n = 298)** | 23 (20–27) | 24 (21–27) | 23 (20–26) | 0.0997 |
| < 15 years | 1 (0.3) | 0 (0) | 1 (0) | 1 |
| 15–45 years | 290 (97.3) | 48 (92) | 242 (98) | |
| > 45 years | 7 (2.3) | 4 (8) | 3 (1) | |
| **Gender (n = 298)** | | | | |
| Male | 296 (99.3) | 52 (100) | 244 (99) | 1 |
| Female | 2 (0.7) | 0 (0) | 2 (1) | |
| **Duration of illness (n = 298), median (IQR)** | 1 (0.7–2) | 1 (0.5–2) | 1.9 (0.7–2) | 0.1493 |
| < 1 month | 94 (31.5) | 21 (40) | 73 (30) | 0.0744 |
| 1–3 months | 159 (53.4) | 28 (54) | 131 (53) | |
| > 3 months | 45 (15.1) | 3 (6) | 42 (17) | |
| **Concomitant Infections (n = 298)** | | | | 0.0104 |
| No | 90 (30.2) | 8 (15) | 82 (33) | |
| Yes | 208 (69.8) | 44 (85) | 164 (67) | |
| Tuberculosis | 65 (31.1) | 13 (30) | 52 (32) | 0.8 |
| Intestinal parasitosis | 62 (29.7) | 11 (25) | 51 (31) | 0.44 |
| Otitis media | 4 (1.9) | 2 (5) | 2 (1) | 0.1958 |
| Pneumonia | 9 (4.3) | 3 (7) | 6 (4) | 0.4014 |
| Others | 69 (33) | 15 (7) | 54 (33) | 0.8643 |
| **VL occurrence (n = 294)** | | | | 0.3135 |
| First episode | 279 (94.9) | 48 (92) | 231 (95) | |
| Relapse | 15 (5.1) | 4 (8) | 11 (5) | |
| **HIV status (n = 275)** | | | | 0.0025 |
| Negative | 261 (93.9) | 39 (83) | 226 (96) | |
| Positive | 17 (6.1) | 8 (17) | 9 (4) | |
| **Type of VL treatment (n = 294)** | | | | |
| SSG + PM | 229 (77.9) | 25 (50) | 204 (84) | <0.001 |
| SSG alone | 2 (0.7) | 0 (0) | 2 (1) | 1 |
| L-AMB | 58 (19.7) | 22 (44) | 36 (15) | <0.001 |
| L-AMB + Miltefosine | 4 (1.4) | 2 (4) | 2 (1) | 0.1355 |
| Other * | 1 (0.3) | 1 (2) | 0 (0) | 0.1701 |
| **VL treatment outcome (n = 295)** | | | | 0.2865 |
| Good | 280 (94.9) | 45 (92) | 235 (96) | 0.2865 |
| Poor | 15 (5.1) | 4 (8) | 11 (4) | |

AKI: acute kidney injury, IQR: Inter Quartile Range, VL: Visceral Leishmaniasis, HIV: Human Immunodeficiency Virus, SSG: Sodium stibogluconate, PM: Paromomycin, L-AMB: Liposomal amphotericin B.

* This patient switched to more than two regimens.

The main etiologies of AKI at admission and during the treatment course were dehydration and drug-induced AKI accounting for 57% and 50% of the patients respectively. Sixty seven percent (20/30) of the patients who had AKI at admission were treated with L-AMB. Out of the 22 patients who developed post-admission AKI, 11 (50%) patients had presumed drug-induced AKI due to SSG-based therapy and of these, six patients completed treatment using the same regimen. The remaining five were shifted to L-AMB due to worsening renal failure (data not shown).

**Table 2. Baseline laboratory characteristics of VL patients at LRTC between January 2019 –December 2019, Gondar, Ethiopia, N = 298.**

| Variables | Total N (%) | AKI | No AKI | P value |
|---|---|---|---|---|
| | | N (%) | N (%) | |
| **WBC, median (IQR) (n = 278)** | 1.4 (1.1–2.0) | 1.42 (1.09–1.9) | 1.4 (1.1–2.0) | 0.999 |
| Leukopenia[a] | 266 (95.7) | 48 (92) | 238 (97) | 0.2336 |
| Normal, n (%) | 12 (4.3) | 4 (8) | 8 (3) | |
| **Hemoglobin median IQR (n = 298)** | 8.3 (6.7–10.8) | 7.9(6.6–9.3) | 8.5 (6.68–9.6) | 0.2395 |
| Anemia[b] | 298 (100) | 52 (100) | 246 (100) | 1 |
| No anemia | 0 | 0 | 0 | |
| **Platelet count, median (IQR) (n = 294)** | 52 (31–85) | 50 (29.5–74.0) | 51 (31–84) | 0.5 |
| Mild thrombocytopenia[c] | 38 (12.9) | 4 (8) | 34 (14) | 0.2545 |
| Moderate thrombocytopenia | 112 (38.1) | 21 (42) | 91 (37) | 0.5326 |
| Severe thrombocytopenia | 132 (44.9) | 22 (44) | 110 (45) | 0.8886 |
| Normal platelets | 12 (4.1) | 3 (6) | 9 (4) | 0.5347 |
| **Proteinuria[d] (n = 225)** | | | | 0.8908 |
| Yes | 128 (56.9) | 22 (58) | 106 (57) | |
| No | 97 (43.1) | 16 (42) | 81(43) | |
| **Hematuria[e] (n = 221)** | | | | 0.0823 |
| Yes | 67(30.3) | 16 (42) | 51(28) | |
| No | 154 (69.7) | 22 (58) | 132 (72) | |
| **Pyuria[f] (n = 220)** | | | | 0.1566 |
| Yes | 90 (40.9) | 19 (51) | 71 (39) | |
| No | 130 (59.1) | 18 (49) | 112 (61) | |
| **Hypoalbuminemia[g] (n = 275)** | | | | 1 |
| Yes | 261 (94.9) | 47 (96) | 214 (95) | |
| No | 14 (5.1) | 2 (4) | 12 (5) | |
| **Hyperbilirubinemia[h] (n = 171)** | | | | 0.3462 |
| Yes | 60 (35.1) | 12 (43) | 48 (34) | |
| No | 111 (64.9) | 16 (57) | 95 (66) | |
| **T-Bil median/IQR (n = 171)** | 1.4 (.9–3.62) | 0.85 (.05–2.48) | 0.85 (.5–1.3) | 0.5476 |
| **D-Bil median/IQR (n = 33)** | 1.2 (.4–2.3) | 1.9 (1.6-.4.5) | 1.1 (.3–2.1) | 0.1452 |
| **Creatinine median/IQR (n = 298)** | 0.9 (0.7–1) | 1.3 (.9–1.7) | 0.8 (.7-.9) | 0.7848 |
| **AST elevation (n = 291)** | | | | 0.4981 |
| Normal | 48 (16.1) | 12 (23) | 36 (15) | |
| 1–3 X ULN | 132 (44.3) | 22 (42) | 110 (45) | |
| 3–5 X ULN | 70 (23.5) | 11 (21) | 59 (24) | |
| >5 ULN | 48 (16.1) | 7 (13) | 41 (17) | |
| **ALT elevation (n = 267)** | | | | |
| Normal | 156 (52.3) | 31 (60) | 125 (51) | 0.6190 |
| 1–3 X ULN | 122 (40.9) | 18 (35) | 104 (42) | 0.8837 |
| 3–5 X ULN | 13 (4.4) | 3 (6) | 10 (4) | 0.6658 |
| >5 X ULN | 7 (2.3) | 0/0 | 7 (3) | 1 |

AKI: acute kidney injury, IQR: Inter Quartile Range, WBC: White blood cells, BilT: Total bilirubin, BilD: Direct bilirubin, AST: Aspartate Aminotransferase, ALT: Alanine Aminotransferase, ULN: Upper Limit of Normal.

a: leukopenia is defined as WBC count <4000.

b: anemia is defined as Hgb<12mg/dl for female and < 13mg/dl for male.

c: thrombocytopenia defined as mild when platelet count s between 100, 000 and 150,000, moderate as a count of 50000 and 100000 and severe if count is less than 50000.

d. proteinuria is defined as urine dipstick for protein ≥ 1+ or 24 hr protein >150mg/day.

e. hematuria is defined as ≥3RBCs/High power field (HPF) on urine microscopy.

f. pyuria is defined as ≥ 5WBC/HPF on urine microscopy.

g: hypoalbuminemia is defined as serum albumin value less than 3.5gm/dl.

h: hyperbilirubinemia is defined as a total bilirubin value increased above the upper normal value, 1.2mg/dl.

**Table 3. Clinical characteristics of VL patients with AKI at LRTC between January 2019 –December 2019, Gondar, Ethiopia, N = 52.**

| Etiology of AKI (n = 52) | N | % |
|---|---|---|
| Pre-renal (dehydration) | 20 | 39 |
| Septic ATN | 9 | 17 |
| SSG+PM induced AKI* | 11 | 21 |
| L-AMB induced AKI* | 1 | 2 |
| Glomerulonephritis | 1 | 2 |
| Not known | 10 | 19 |
| **Stage of AKI at diagnosis (n = 52)** | | |
| Stage 1 | 34 | 66 |
| Stage 2 | 10 | 19 |
| Stage 3 | 8 | 15 |
| **Active urinary sediments** | | |
| **Proteinuria (n = 38)** | | |
| Yes | 22 | 58 |
| No | 16 | 42 |
| **Hematuria (n = 38)** | | |
| Yes | 16 | 42 |
| No | 22 | 58 |
| **Pyuria (n = 37)** | | |
| Yes | 19 | 51 |
| No | 18 | 49 |
| **Outcome of AKI (n = 52)** | | |
| Resolved | 40 | 77 |
| Not Resolved | 5 | 9.6 |
| Died | 2 | 3.8 |
| Unknown | 5 | 9.6 |

AKI: Acute Kidney Injury, ATN: Acute Tubular Necrosis, L-AMB: liposomal amphotericin B.

*Drug induced AKI occurs post-admission.

The majority (40/52) of the patients with AKI had good renal outcome, with normalization of renal function. However, 12 patients had a bad renal outcome with two deaths, five AKI that had not resolved by the time of discharge or end of VL treatment. In the remaining five, the AKI status was unknown at the end of the VL treatment. One patient died of respiratory

**Table 4. Rate of post-admission AKI per initial treatment allocation among patients with VL with baseline normal kidney function at LRTC between January 2019 –December 2019, Gondar, Ethiopia, N = 268.**

| Type of treatment | Post-admission AKI (n = 265*) | | OR | p value |
|---|---|---|---|---|
| | No | Yes | | |
| SSG-based[a] | 206 (93%) | 15 (7%) | 1 | 0.13 |
| L-AMB based[b] | 38 (86%) | 6 (14%) | 2.17 (0.8–6.0) | |

AKI: acute kidney injury, SSG: Sodium stibogluconate, PM: Paromomycin, L-AMB: liposomal amphotericin B; OR: odds ratio.

*Data missing for 3 patients.

[a]SSG-based: SSG+ PM (n = 219), SSG alone (n = 2).

[b]L-AMB-based: L-AMB alone (n = 40), L-AMB + Miltefosine (n = 4).

failure due to pneumonia and the other patient died of septic shock. In both cases, AKI had not resolved at the time of death (**Table 3**).

## Discussion

This is one of the first studies from sub-Saharan Africa assessing the prevalence of AKI and other renal clinical manifestations in VL patients in Ethiopia. Renal involvement in VL has been reported before from outside of Africa in the form of urinary abnormalities, AKI and histological abnormalities in kidney biopsy [9, 25]. A total of 298 patients were included in the analysis and the spectrum of renal diseases included AKI (17.4%), proteinuria (56.9%), hematuria (30.3%) and pyuria (40.9%). The observed proportion of AKI (17.4%) was quite different from the assumed proportion (50%) used to calculate the sample size.

The proportion of AKI in patients with VL in our study was 17.4% (52/298). Our findings are similar to previous reports from India (15%) but an AKI prevalence of 26–37% has been reported from Brazil [25–27]. The reported incidence of AKI in VL varied from 4.2 to 45.9% [7, 10, 28]. The use of different criteria to define AKI and differences in etiological factors causing AKI may contribute to this variation in incidence of AKI in patients with VL [25].

In our study, proteinuria was found in 56.9% of VL patients using a semi-quantitative urine dipstick test. Microscopic hematuria and pyuria were observed in 30.3% and 40.9% of VL patients respectively but there was no difference in the occurrence of proteinuria and hematuria in patients with or without AKI (p > 0.05). This finding is consistent with previous studies. Abnormal urinary sediments (microscopic hematuria and proteinuria) in VL was observed by several investigators [18, 19]. A prospective study in Brazil in VL patients demonstrated hematuria, mild to moderate proteinuria, and increased urine leukocytes in over 50% of cases [29]. Another study from India, reported proteinuria ranging between 1.2–1.8gm/24 hrs in 15% of cases [25]. Proteinuria was also observed in patients with normal SCr and this is consistent with prior reports where proteinuria was detected in more than 40% of VL patients, even in those with normal SCr levels [30]. Urinary protein excretion in VL patients is usually in the range of <1 g/24 h [10]. However, as proteinuria was not measured quantitatively in our study, direct comparisons of our findings with these studies is not possible.

The most common cause of AKI at admission was pre-renal azotemia (fluid loss from diarrhea and vomiting, poor fluid intake). AKI in VL patients in previous reports was also mostly related to pre-renal factors [31]. Sepsis and drug induced AKI were the second most common causes of AKI in the current study. The first line of treatment for clinically stable HIV negative patients with VL in the study area is SSG + PM and most patients received this regimen while patients who are critically ill, having HIV-confection and significant renal and/or hepatic dysfunction received L-AMB.

The rate of post-admission AKI was 8.2% (22/268), of which eleven patients developed AKI due to presumed SSG + PM renal toxicity while only one patient had AKI due to presumed L-AMB renal toxicity (**Table 3**). However there was no statistically significant difference in the rate of post-admission AKI among patients treated with SSG-based and L-AMB based treatment (**Table 4**). Even though statistically not significant, patients who were treated with L-AMB regimen seems more likely to develop AKI compared with those treated with SSG-based regimen(OR = 2.17 CI:0.8–6.0). This is likely because patients who received L-AMB were either critically ill or have HIV co-infection and this will increase the risk of AKI independent of the anti-leishmanial treatment. Both SSG and PM are potentially nephrotoxic and AKI related to these drugs has been reported [31]. Previous clinical trials on safety and effectiveness of SSG and PM in East Africa have documented renal failure with fatality as severe adverse events [14]. L-AMB is generally very safe, however, increased serum urea and SCr due

to amphotericin B have been reported [7, 9, 16]. One patient in our study developed AKI due to glomerulonephritis. This patient had renal failure with hypertension, active urine sediments, fluid overload and massive proteinuria of 6500mg/24hrs. Similar to this case, VL related glomerulonephritis has been reported [18, 32, 33].

In previous studies, several factors have been identified as risk factors for the development of AKI in VL patients including male gender, advanced age, jaundice, hypokalemia, leukopenia, use of conventional amphotericin B and secondary infections [8, 27]. 99.3% of the study participants in this study were male and this is consistent with prior reports of male predominance of 96–100% in the study area [22, 34]. This very significant male predominance could be due to presence of male seasonal daily laborers in the study area. In our study, a total of 17 patients (6.1%) had VL-HIV co-infection and HIV co-infection was an independent risk factor for the development of AKI. VL patients with HIV co-infection were 6 (AOR = 6.01 95% CI: 1.99–18.27, p = 0.001) times more likely to develop AKI compared to HIV negative VL patients VL is an important opportunistic disease in HIV-positive patients in Ethiopia. The Northwest part of Ethiopia, where this study was conducted, has the highest HIV-VL co-infection rate globally [34, 35]. The reasons for the increased incidence of AKI in HIV-VL co-infection could be multiple, including the occurrence of more severe VL, HIV per se, concomitant opportunistic infections or medication. There have been very few case reports of HIV-VL co-infected patients who developed renal complications. Reported renal manifestations in this group of patients included glomerular disease, nephrotic syndrome, AKI, and amyloidosis [13, 17, 20, 21].

In line with other studies, AKI also tended to occur more commonly in patients with concomitant secondary infections. Patients with concomitant secondary infections were 3.44 (AOR = 3.44 95% CI: 1.37–8.65, p = 0.009) times more likely to develop AKI than patients without secondary infections. Similarly to others, we also observed that AKI was reversible with correction of dehydration, treatment of concomitant bacterial infections and treatment of VL in the majority of cases [28, 31]. Additionally, mortality was low in this study. There were two fatalities in this study due to respiratory failure and refractory shock. Presence of AKI had likely contributed to the death in both patients. Routine monitoring of renal function is recommended to detect renal complications early on and guide patient management including fluid management and the choice of VL treatment.

There are several limitations to this study. Given the retrospective nature, there were some missing data, especially dipstick and urine microscopy were not done for all patients. Proteinuria was tested semi-quantitatively using urine dipstick; 24 hr urine protein excretion would have been more useful. Renal biopsy was not performed in those patients with features of glomerular disease, and hence it is difficult to characterize the nature of glomerular and tubular lesions. The etiology of AKI was based on the judgment of the treating physician and establishing the causality of AKI might be inaccurate.

## Conclusion

In summary, renal disease occurred at the time of admission and during the course of treatment of VL patients. The spectrum of renal disease included AKI, proteinuria, hematuria, and pyuria. Presence of HIV co-infection and other secondary infections were independent risk factors for the development of AKI. The majority of the renal manifestations were generally of mild nature and reversible with treatment of VL, rehydration and treatment of bacterial co-infections without specific treatment. Further study is required to address the degree of proteinuria quantitatively, the nature of the glomerular and tubular lesions, the influence of AKI in the treatment course and its contribution for morbidity and mortality in patients with VL with and without HIV co-infection.

## Acknowledgments

We thank all LRTC staff members who are actively involved in the management of these patients. Our acknowledgement also goes to the Drugs for Neglected Disease Initiative (DNDi) and University of Gondar (UoG) for supporting the LRTC.

## Author Contributions

**Conceptualization:** Workagegnehu Hailu, Rezika Mohamed, Helina Fikre, Saba Atnafu, Azeb Tadesse, Ermias Diro, Johan van Grienvsen.

**Data curation:** Workagegnehu Hailu, Rezika Mohamed, Saba Atnafu, Azeb Tadesse.

**Formal analysis:** Workagegnehu Hailu, Rezika Mohamed, Ermias Diro, Johan van Grienvsen.

**Methodology:** Workagegnehu Hailu, Rezika Mohamed, Helina Fikre, Ermias Diro, Johan van Grienvsen.

**Project administration:** Workagegnehu Hailu.

**Supervision:** Rezika Mohamed, Ermias Diro, Johan van Grienvsen.

**Writing – original draft:** Workagegnehu Hailu, Rezika Mohamed.

**Writing – review & editing:** Workagegnehu Hailu, Rezika Mohamed, Helina Fikre, Saba Atnafu, Azeb Tadesse, Ermias Diro, Johan van Grienvsen.

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
