## [Decision Letter · Decision Letter 0]

17 Feb 2021

PONE-D-21-00134

Acute Kidney Injury in patients with Visceral Leishmaniasis in Northwest Ethiopia

PLOS ONE

Dear Dr. Hailu,

Thank you for submitting your manuscript to PLOS ONE. After careful consideration, we feel that it has merit but does not fully meet PLOS ONE’s publication criteria as it currently stands. Therefore, we invite you to submit a revised version of the manuscript that addresses the points raised during the review process.

I enjoyed reading your manuscript and believe the reviewers have provided a number of constructive comments to improve clarity and help strengthen the manuscript. 

We look forward to receiving your revised manuscript.

Kind regards,

Andrea L. Conroy, PhD

Academic Editor

PLOS ONE

Journal Requirements:

2. Thank you for stating in your ethics statement "As a retrospective study patient informed consent is waivered." Please clarify whether your ethics board specifically granted a waiver of informed consent. Please add this information to your manuscript and to the ethics statement in the online submission form.

3. Thank you for providing the date(s) when patient medical information was initially recorded. Please also include the date(s) on which your research team accessed/retrieved the medical records data used in your study.

"Our acknowledgement also goes to the Drugs for Neglected Disease Initiative (DNDI)

and University of Gondar (UoG) for supporting the Leishmaniasis Research and Treatment

Canter."

"The authors received no specific funding for this work"

6. Please amend your list of authors on the manuscript to ensure that each author is linked to an affiliation. Authors’ affiliations should reflect the institution where the work was done (if authors moved subsequently, you can also list the new affiliation stating “current affiliation:….” as necessary).

7. We noticed you have some minor occurrence of overlapping text with the following previous publication(s), which needs to be addressed:

- https://www.cell.com/heliyon/fulltext/S2405-8440(18)31817-6?_returnURL=https%3A%2F%2Flinkinghub.elsevier.com%2Fretrieve%2Fpii%2FS2405844018318176%3Fshowall%3Dtrue

- https://apps.who.int/iris/bitstream/handle/10665/311922/IC_Meet_Rep_2019_EN_20619.pdf

- https://cdn.intechopen.com/pdfs-wm/31093.pdf

- http://ispub.com/journal/the_internet_journal_of_tropical_medicine/volume_4_number_1_51/article/spectrum_of_renal_disease_in_visceral_leishmaniasis .html?

In your revision ensure you cite all your sources (including your own works), and quote or rephrase any duplicated text outside the methods section. Further consideration is dependent on these concerns being addressed.

Reviewers' comments:

Reviewer's Responses to Questions

**Comments to the Author**

1. Is the manuscript technically sound, and do the data support the conclusions?

Reviewer #1: Partly

Reviewer #2: Yes

2. Has the statistical analysis been performed appropriately and rigorously? 

Reviewer #1: No

Reviewer #2: No

3. Have the authors made all data underlying the findings in their manuscript fully available?

Reviewer #1: No

Reviewer #2: Yes

4. Is the manuscript presented in an intelligible fashion and written in standard English?

Reviewer #1: No

Reviewer #2: Yes

5. Review Comments to the Author

Reviewer #1: Thank you for this study on AKI in Leishmaniasis; a topic which has limited data. Below are my comments in regard to the manuscript.

1) In the conclusion of the abstract, you report'....proteinuria, hematuria, pyuria and glomerulonephritis. How did you separate glomerulonephritis from hematuria and proteinuria.' Are these not presentations which made you diagnose those patients with glomerulonephritis? You need to review the statement and clarify how glomerulonephritis patients differed from those reported to have proteinuria and hematuria.

2) In the introduction you state'....Visceral Leishmaniasis (VL) is the most serious form of the disease with 100% mortality if left untreated.' May you provide a reference for the statement

3) In the introduction, you wrote '.....In Ethiopia, where SSG and PM combination is used as a first line treatment

and where there is a high rate of HIV'. However, PM abbreviation was not written in full. May you write in full on first use of the abbreviations

4) Under methods section you stated '....Patients included in ongoing clinical trials during this period, charts with significantly incomplete data or patient files that could not be retrieved at LRTC chart room were excluded. (22).' There is need to improve on the gramma in this sentence. It isn't clear in its current form.

5) When defining AKI, it is unclear what creatinine measurement was used to define baseline creatinine. This needs to be specified clearly

6) In the results you write...A total of 17 patients (6.1%) had VL-HIV co-infection, of which alf of the cases (53%; 9/17) were on antiretroviral. There is an error in this statement, it could be that the word 'alf' was to mean half

7) In table 1, specify which drug for VL is 'Other' since it was only used by one person

8) In general, there are a number of grammatical errors, there is need to improve the manuscript by ensuring that the gramma is improved.

9) Risk factors were based on a bivariate table 1. There was need for multivariate analysis to provide better results

10) It is unclear how AKI at admission was defined and how this definition differed from AKI during treatment.

There is also lack of clarity on how dehydration was determined and what was meant by nephrotoxicity; and what nephrotoxic drugs were being referred to or evaluated.

11) Table 3 and Table 4 provide etiology of AKI; from the study design it maybe hard to provide evidence of causality of AKI by these factors

12) In the conclusion section the researcher writes; 'In summary, renal disease occurred before VL diagnosis and during the course of treatment'. However, this may not be entirely based on the results since the study was evaluating patients admitted to hospital and not before diagnosis. The study doesn't evaluate patient before VL diagnosis but provides data on patients who had been managed for VL at the VL treatment centre. There is need to make more clarification on this and the conclusion from the study.

In addition, the researcher states in the conclusion that the renal manisfestations were mild in nature and reversible; however, there was stage 3 AKI and the research never provided recovery data to support the statement of reversibility

Reviewer #2: The authors are to be congratulated on this interesting report of AKI in VL from Ethiopia.

The causes of AKI at admission (present in 10% of patients) included dehydration. HIV and other co-infections were risk factors for AKI (interesting).

AKI also developed in 22 patients who did not have AKI at admission, which could be related to medications.

This manuscript warrants eventual publication, but I recommend some revisions to improve the clarity of the findings reported. As outlined below, the reporting of possible drug-associated AKI is inadequate and possibly even misleading.

Major revision: improved reporting of possible drug-associated AKI

AKI is multi-factorial in this patient cohort.

The authors have done well to separate AKI at admission from AKI during treatment to tease out the effect of medications, for example. Can the authors please confirm (and report in the revised manuscript methods and/or results), whether patients referred to Gondar Hospital had any prior treatment for their VL or other conditions at other centres? Prior treatment with L-AMB or pentavalent antimonials or aminoglycosides/vancomycin/ other nephrotoxic antibiotics. It’s not explicitly stated whether the patients were treatment experienced (potential exposure to nephrotoxins) or treatment naïve. If treatment naïve, this is the “natural history” of AKI in VL as opposed to iatrogenic. As a referral centre, it is possible that patients had been previously hospitalized and treated elsewhere with or without an appropriate diagnosis of VL, for example.

The development of incident AKI after admission in 22 patients may be related to medications (SSG or L-AMB). I strongly recommend that the rate of post-admission AKI be expressed as the proportion of those without AKI at admission who subsequently develop AKI (authors are using the wrong denominator when they divide by the whole cohort n=298):

I believe there should be 268 patients without AKI at admission (298-30):

22/268 = 8.2% (denominator is those without AKI at admission).

I recommend that the rate be compared in those treated with L-AMB and those treated with SSG:

Of the 268 patients without AKI at admission, how many were treated with L-AMB, how many with SSG?

What was the rate of AKI in the SSG group? (excluding patients with AKI at admission)

What was the rate of AKI in the L-AMB group? (excluding patients with AKI at admission)

Could consider a statistical comparison.

If these data can be deduced from the tables 1 and 4, I get the following (excluding patients with AKI at admission):

Drug no AKI post-admission AKI

SSG 204 19

L-AMB 38 3

Total: 242 22 Overall total: 264 (approx. 268, excluding AKI at admission)

Rate of AKI in SSG group: 19/223 = 8.5%

Rate of AKI in L-AMB exposed: 3/41 = 7.3%

OR = 1.16, p-value >0.99

This level of detail is essential to understand the effects of the medications on incident AKI after treatment initiation.

The current reporting is not sufficiently detailed and even misleading:

“Most (67%; 20/30) patients who had AKI at admission were treated with AmBisomeTM whereas 86% (19/22) of patients who develop AKI during VL treatment course were on SSG +PM”

After initial read, I though the SSG was associated with AKI, but a more careful review of the data suggests to me there is no evidence of higher rates of incident AKI in the SSG-treated patients.

This is partially addressed in Table 4. However, with respect to drug treatment, it would be helpful to add to table 4 those patients who did not have AKI at admission or during treatment (n=246, control group without AKI at admission, who did not develop AKI subsequently)– which drugs received and whether the drugs were switched. This could be included as a third column in Table 4 (I realize there wouldn’t be entries for “etiology of AKI” in this group!). This is an essential comparator group to make sense of the treatment numbers.

In summary, the reporting of AKI after admission according to treatment is not sufficiently detailed (Major revision necessary since this is a central mechanism of the AKI in this study).

Minor revisions:

Abstract:

17.4% (52/298) VL patients had AKI: recommend giving the 95% confidence interval on this proportion, since it’s the primary objective of the study.

AmBisome: liposomal amphotericin B. Please use generic rather than trade name

Introduction:

Glucantime: meglumine antimoniate. please use generic rather than trade name. (Introduction, 3rd paragraph)

PM: paromomycin? Please spell out the name with abbreviation when first mentioned (Introduction, 3rd paragraph)

Methods:

“all patients admitted during the study period at LRTC with confirmed diagnosis of VL” – should state the way the diagnosis was confirmed (e.g., confirmed by the presence of amastigotes on splenic or bone marrow aspirate - Giemsa-stain, light microscopy…) Please explicitly state if serology, PCR, and culture were not used for diagnosis.

Sample size calculation:

First, the authors are to be congratulated on including a sample size calculation for this observational study (frequently omitted).

I note that the authors used the alpha=90% level of significance for this calculation (z-score[0.05]=1.64):

n=(p)*(1-p)*(1.64/d)^2 = (0.5)*(1-0.5)*(1.64/0.05)^2 = 270

whereas alpha=95%CI is more standard (z-score[0.025]=1.96):

n=(p)*(1-p)*(1.96/d)^2 = (0.5)*(1-0.5)*(1.96/0.05)^2 = 378

In the end, the estimate of the proportion (50%) is quite different from the observed proportion (17%) and the confidence interval on the proportion is still within +/-5%

Overall, I think the sample size is adequate, but the assumptions of the sample size calculation are different from convention and from the observed proportion. A brief comment in the discussion on the adequacy of the sample size and differences between a priori assumed rate and observed rate may be considered.

The sampling strategy for the chart review is also clear and I congratulate the authors for describing this.

Results.

All but two of the charts reviewed were of male patients?

This is surprising. Although there is a male predominance in VL, it is far from an exclusively male disease. (e.g., male/female ratio was 1.40 in PLOS NTD, Male predominance in reported Visceral Leishmaniasis cases: Nature or nurture? A comparison of population-based with health facility-reported data.)

The ratio in this study would be 296:2 !

I think this requires an explanation why female patients were not included (not likely chance alone).

Risk factors for AKI (Co-infection and HIV): please give odds ratio and 95%CI for the association between HIV and AKI, likewise for co-infections and AKI.

Editorial/language revisions:

clinician judgment of the treating physician: clinical judgment of the treating physician (Methods, Variables and operational definitions, paragraph 2)

Patient’s data confidentially were respected: patient data confidentiality was respected (Methods, ethical considerations)

“The main etiology of AKI at admission and during treatment course were dehydration and nephrotoxicity” – "nephrotoxicity" is not really an etiology (too general a term) – perhaps reword: medication-induced tubular injury or glomerular injury, for example. (Results)

Table 2: AST elevation. LN (missing U for ULN)

6. PLOS authors have the option to publish the peer review history of their article (what does this mean?). If published, this will include your full peer review and any attached files.

Reviewer #1: No

Reviewer #2: **Yes: **Michael Hawkes

---

## [Author Response · Author response to Decision Letter 0]

24 Apr 2021

1. Response to the academic editor’s comments

Journal Requirements:

Reply: Thank you for the comments. We have edited both the authors’ affiliation and the main manuscript body as per the PLOS ONE style requirement. 

2. Thank you for stating in your ethics statement "As a retrospective study patient informed consent is waivered." Please clarify whether your ethics board specifically granted a waiver of informed consent. Please add this information to your manuscript and to the ethics statement in the online submission form.

Reply: Yes, the ethics board has granted a waiver of informed consent and we have added this information to the manuscript and in the online submission form. Line #175-176

3. Thank you for providing the date(s) when patient medical information was initially recorded. Please also include the date(s) on which your research team accessed/retrieved the medical records data used in your study.

Reply: This is a relevant remark. Data was retrieved from the medical records over a period of 2 weeks (Feb 3-15, 2020). We have included this in the “methods” part, line # 107-108. 

"Our acknowledgement also goes to the Drugs for Neglected Disease Initiative (DNDI)

and University of Gondar (UoG) for supporting the Leishmaniasis Research and Treatment

Canter."

"The authors received no specific funding for this work"

Reply: We haven’t got any specific fund for this work. We retrieved the data from LRTC. This center is funded by DNDi. That is why we put DNDI on the acknowledgment part. 

Reply: The ethical statement is moved to the methods part and removed from the declaration section. Line #175-179

6. Please amend your list of authors on the manuscript to ensure that each author is linked to an affiliation. Authors’ affiliations should reflect the institution where the work was done (if authors moved subsequently, you can also list the new affiliation stating “current affiliation:….” as necessary).

Reply: All authors’ affiliation is to University of Gondar where the work has been done EXCEPT for Johan van Grienvsen whose affiliation is to ITM, Belgium. Even though he is not a member of the University of Gondar, his contribution to this manuscript is significant to be a co-author. 

7. We noticed you have some minor occurrence of overlapping text with the following previous publication(s), which needs to be addressed:

- https://www.cell.com/heliyon/fulltext/S2405-8440(18)31817-6?_returnURL=https%3A%2F%2Flinkinghub.elsevier.com%2Fretrieve%2Fpii%2FS2405844018318176%3Fshowall%3Dtrue

- https://apps.who.int/iris/bitstream/handle/10665/311922/IC_Meet_Rep_2019_EN_20619.pdf

- https://cdn.intechopen.com/pdfs-wm/31093.pdf

- http://ispub.com/journal/the_internet_journal_of_tropical_medicine/volume_4_number_1_51/article/spectrum_of_renal_disease_in_visceral_leishmaniasis .html?

In your revision ensure you cite all your sources (including your own works), and quote or rephrase any duplicated text outside the methods section. Further consideration is dependent on these concerns being addressed.

Reply: Thank you. We have rephrased and used our own words to avoid overlapping texts with the above mentioned 4 previous publications that we cited. Line #71, 72, 77, 85

2. Answer to the reviewer’s comments 

Reviewers' comments:

Reviewer #1: Thank you for this study on AKI in Leishmaniasis; a topic which has limited data. Below are my comments in regard to the manuscript.

Reply: Thanks for the positive feedback

1) In the conclusion of the abstract, you report'....proteinuria, hematuria, pyuria and glomerulonephritis. How did you separate glomerulonephritis from hematuria and proteinuria.' Are these not presentations which made you diagnose those patients with glomerulonephritis? You need to review the statement and clarify how glomerulonephritis patients differed from those reported to have proteinuria and hematuria.

Reply: The question is indeed relevant. Yes proteinuria and hematuria are the cardinal features of glomerulonephritis. In this study, all but one of the patients had either proteinuria or hematuria without having other clinical features of glomerulonephritis (edema, hypertension and renal insufficiency). The causes of these urinary findings could be fever, sepsis or glomerulonephritis (but we haven’t performed biopsy to confirm this). There was only one patient who had anasarca, massive proteinuria, hypertension and renal failure fulfilling all the clinical criteria for glomerulonephritis. We have removed glomerulonephritis from the conclusion part of the statement (as there is just one case). Line # 54 and 401

2) In the introduction you state'....Visceral Leishmaniasis (VL) is the most serious form of the disease with 100% mortality if left untreated.' May you provide a reference for the statement

Reply: While searching prognosis of untreated VL, we found it is around 95% fatality (WHO fact sheet). We have adjusted this statement accordingly and included the reference. Line# 75

3) In the introduction, you wrote '.....In Ethiopia, where SSG and PM combination is used as a first line treatment

and where there is a high rate of HIV'. However, PM abbreviation was not written in full. May you write in full on first use of the abbreviations

Reply: Thanks, corrected accordingly. Line #99

4) Under methods section you stated '....Patients included in ongoing clinical trials during this period, charts with significantly incomplete data or patient files that could not be retrieved at LRTC chart room were excluded. (22).' There is need to improve on the gramma in this sentence. It isn't clear in its current form.

Reply: we have edited the statement like this” Exclusion criteria were patients participating in ongoing clinical trials, if serum creatinine was not measured and if patient files could not be retrieved at the LRTC chart room” Line# 128-132.

5) When defining AKI, it is unclear what creatinine measurement was used to define baseline creatinine. This needs to be specified clearly

Reply: We used the KDIGO definition as it is. For patients who developed AKI after admission, we use the initial creatinine determined in the hospital as the baseline creatinine. But for those who had elevated creatinine from the outset (at admission), and if it is presumed to occur within the last 7 days, we take it as AKI (that is why we mentioned “Known or presumed” to occur within the past 7 days on the definition part. This is true also in clinical practice that most patients with AKI do not have baseline creatinine before admission, but if they present with risk factors that can cause AKI, the elevated creatinine will be assumed to be of new onset. 

6) In the results you write...A total of 17 patients (6.1%) had VL-HIV co-infection, of which alf of the cases (53%; 9/17) were on antiretroviral. There is an error in this statement, it could be that the word 'alf' was to mean half

Reply: yes, this is typo (we mean”half”) - corrected. Line # 194

7) In table 1, specify which drug for VL is 'Other' since it was only used by one person

Reply: This patient switched to more than two regimens. Hence it was difficult to put a single treatment allocation. We put a foot note under table 1 to describe this. 

8) In general, there are a number of grammatical errors, there is need to improve the manuscript by ensuring that the gramma is improved.

Reply: Thanks, we have tried to improve the grammatical errors in the revised manuscript.

9) Risk factors were based on a bivariate table 1. There was need for multivariate analysis to provide better results

 Reply: This is a valid comment. We run a multivariate analysis and we found that HIV status and concomitant infections were factors that were significantly associated with development of AKI. We have provided this information on the result section. Line # 245-249. The abstract is revised according to this finding. Line# 48-50

10) It is unclear how AKI at admission was defined and how this definition differed from AKI during treatment.

Reply: As we tried to explain above, AKI at admission was an elevated serum creatinine that was presumed to occur recently (within 7 days) - see definition. While AKI after admission was an elevation in creatinine level from the baseline creatinine that was determined at admission. 

There is also lack of clarity on how dehydration was determined and what was meant by nephrotoxicity; and what nephrotoxic drugs were being referred to or evaluated.

Reply: These possible causes of AKI were assigned by the treating physician (pre-renal/dehydration- those patients having AKI and, history of fluid loss from diarrhea, vomiting with low blood pressure), drug induced AKI is defined if a patient develop AKI that is presumed due to the VL treatment drugs (excluding sepsis and pre-renal causes). The drugs referred here are SSG, PM and AmBisome (Table 3). We changed the term nephrotoxicity to drug induced AKI (as nephrotoxicity is a very broad term)

11) Table 3 and Table 4 provide etiology of AKI; from the study design it maybe hard to provide evidence of causality of AKI by these factors

Reply: We understand this concern. It may be difficult to provide of causality with the method we used. We used the treating physician’s judgment as a cause of the AKI (mentioned in the operational definitions). This might be inaccurate. In fact AKI etiology is usually obtained clinically (from history, physical examination and lab data) in routine clinical practice. Biopsy is rarely indicated to find etiology of AKI (glomerulonephritis and interstitial nephritis). Nephrotoxicity (considering the time relationship) and pre-renal AKI are usually clinically diagnosed. But we acknowledge your concern and we mentioned this in the limitation part of the revised manuscript. Line # 404-405

We have also amended table 4 as per a comment from the other reviewer. 

12) In the conclusion section the researcher writes; 'In summary, renal disease occurred before VL diagnosis and during the course of treatment'. However, this may not be entirely based on the results since the study was evaluating patients admitted to hospital and not before diagnosis. The study doesn't evaluate patient before VL diagnosis but provides data on patients who had been managed for VL at the VL treatment centre. There is need to make more clarification on this and the conclusion from the study.

Reply: This point is indeed relevant. Yes we agree that, we have seen the patients with VL diagnosis at admission and during follow up. However, for the vast majority of the patients, diagnosis of VL was made at the center after self/health facility referral. We have corrected this statement accordingly as” In summary, renal disease occurred at the time of admission and during the course of treatment of VL patients”. Line # 407-408

In addition, the researcher states in the conclusion that the renal manisfestations were mild in nature and reversible; however, there was stage 3 AKI and the research never provided recovery data to support the statement of reversibility

Reply: 77% of all the AKI have resolved (Table 3). Yes, there were 8 patients with stage 3 AKI, but we didn’t follow outcome specifically for the stage 3. We corrected this statement as ”The majority of the renal manifestations were generally of mild nature and reversible with treatment of VL, rehydration and treatment of bacterial co-infections without specific treatment in the majority of cases” Line # 412

Reviewer #2: The authors are to be congratulated on this interesting report of AKI in VL from Ethiopia.

The causes of AKI at admission (present in 10% of patients) included dehydration. HIV and other co-infections were risk factors for AKI (interesting).

AKI also developed in 22 patients who did not have AKI at admission, which could be related to medications.

This manuscript warrants eventual publication, but I recommend some revisions to improve the clarity of the findings reported. As outlined below, the reporting of possible drug-associated AKI is inadequate and possibly even misleading.

Reply: Thanks for the positive feedback 

Major revision: improved reporting of possible drug-associated AKI

AKI is multi-factorial in this patient cohort.

The authors have done well to separate AKI at admission from AKI during treatment to tease out the effect of medications, for example. Can the authors please confirm (and report in the revised manuscript methods and/or results), whether patients referred to Gondar Hospital had any prior treatment for their VL or other conditions at other centres? Prior treatment with L-AMB or pentavalent antimonials or aminoglycosides/vancomycin/ other nephrotoxic antibiotics. It’s not explicitly stated whether the patients were treatment experienced (potential exposure to nephrotoxins) or treatment naïve. If treatment naïve, this is the “natural history” of AKI in VL as opposed to iatrogenic. As a referral centre, it is possible that patients had been previously hospitalized and treated elsewhere with or without an appropriate diagnosis of VL, for example.

Reply: These points are indeed relevant. Regarding the AKI at admission, it is likely due to the natural history. All patients were treatment naïve. Prior treatment with anti-leishmanial drug or other nephrotoxic medications was not given. These are patients who visited the hospital by self-referral or referred from health centers in the vicinity. We have mentioned this in the result section (baseline demographic and clinical characteristics). Line # 188-190

The development of incident AKI after admission in 22 patients may be related to medications (SSG or L-AMB). I strongly recommend that the rate of post-admission AKI be expressed as the proportion of those without AKI at admission who subsequently develop AKI (authors are using the wrong denominator when they divide by the whole cohort n=298):

I believe there should be 268 patients without AKI at admission (298-30):

22/268 = 8.2% (denominator is those without AKI at admission).

Reply: This is absolutely true. We have corrected the rate of post-admission AKI using the 268 patients as a denominator. Line # 234-235. 

I recommend that the rate be compared in those treated with L-AMB and those treated with SSG:

Of the 268 patients without AKI at admission, how many were treated with L-AMB, how many with SSG?

What was the rate of AKI in the SSG group? (excluding patients with AKI at admission)

What was the rate of AKI in the L-AMB group? (excluding patients with AKI at admission)

Could consider a statistical comparison.

If these data can be deduced from the tables 1 and 4, I get the following (excluding patients with AKI at admission):

Drug no AKI post-admission AKI

SSG 204 19

L-AMB 38 3

Total: 242 22 Overall total: 264 (approx. 268, excluding AKI at admission)

Rate of AKI in SSG group: 19/223 = 8.5%

Rate of AKI in L-AMB exposed: 3/41 = 7.3%

OR = 1.16, p-value >0.99

This level of detail is essential to understand the effects of the medications on incident AKI after treatment initiation.

The current reporting is not sufficiently detailed and even misleading:

“Most (67%; 20/30) patients who had AKI at admission were treated with AmBisomeTM whereas 86% (19/22) of patients who develop AKI during VL treatment course were on SSG +PM”

After initial read, I though the SSG was associated with AKI, but a more careful review of the data suggests to me there is no evidence of higher rates of incident AKI in the SSG-treated patients.

This is partially addressed in Table 4. However, with respect to drug treatment, it would be helpful to add to table 4 those patients who did not have AKI at admission or during treatment (n=246, control group without AKI at admission, who did not develop AKI subsequently)– which drugs received and whether the drugs were switched. This could be included as a third column in Table 4 (I realize there wouldn’t be entries for “etiology of AKI” in this group!). This is an essential comparator group to make sense of the treatment numbers.

In summary, the reporting of AKI after admission according to treatment is not sufficiently detailed (Major revision necessary since this is a central mechanism of the AKI in this study).

Reply: All the raised points are indeed relevant. Thank you for the detailed explanation on how to do the analysis for the drug induced AKI. We just took the frequency of patients who took SSG or L-AMB only from those who develop AKI, and it ends up with a wrong conclusion. According to your comment, we ran the data for the 268 patients, to look at AKI development per treatment arm (we see exposure-SSG or L-AMB) and looked for the rate of outcome (AKI) per treatment allocation. We have reported this in the new Table 4 (we totally changed table 4). Table 3 is also slightly modified. 

As you mentioned above, there is no evidence supporting higher rates of AKI in SSG than L-AMB. Rather the trend seems patients on L-AMB likely to develop AKI than those on SSG based. This is probably because patients who are assigned to L-AMB from the beginning are those who are either critically ill or have HIV co-infection. Such patients are at risk of developing AKI and they were put on L-AMB for fear of SSG toxicity. We mentioned this in the discussion part. Line # 357-363

Minor revisions:

Abstract:

17.4% (52/298) VL patients had AKI: recommend giving the 95% confidence interval on this proportion, since it’s the primary objective of the study.

Reply: A relevant remark. We have included 95% CI. Line # 39-40

AmBisome: liposomal amphotericin B. Please use generic rather than trade name

Reply: we have corrected it accordingly 

Introduction:

Glucantime: meglumine antimoniate. please use generic rather than trade name. (Introduction, 3rd paragraph)

Reply: Corrected 

PM: paromomycin? Please spell out the name with abbreviation when first mentioned (Introduction, 3rd paragraph)

Reply: corrected . Line # 99

Methods:

“all patients admitted during the study period at LRTC with confirmed diagnosis of VL” – should state the way the diagnosis was confirmed (e.g., confirmed by the presence of amastigotes on splenic or bone marrow aspirate - Giemsa-stain, light microscopy…) Please explicitly state if serology, PCR, and culture were not used for diagnosis.

Reply: Majority of patients have a parasitological diagnosis of VL. However, there were patients diagnosed with clinical and serologic criteria. We have adjusted the statement like” Medical charts of all patients admitted during the study period at LRTC with diagnosis of VL were retrieved. Diagnosis of VL was made by either the presence of amastigotes on splenic or bone marrow aspirate - Giemsa-stain under light microscopy or a patient fulfilling WHO case definition for VL and positive rK-39 test if tissue aspiration was not done” Line # 126-128

Sample size calculation:

First, the authors are to be congratulated on including a sample size calculation for this observational study (frequently omitted).

I note that the authors used the alpha=90% level of significance for this calculation (z-score[0.05]=1.64):

n=(p)*(1-p)*(1.64/d)^2 = (0.5)*(1-0.5)*(1.64/0.05)^2 = 270

whereas alpha=95%CI is more standard (z-score[0.025]=1.96):

n=(p)*(1-p)*(1.96/d)^2 = (0.5)*(1-0.5)*(1.96/0.05)^2 = 378

In the end, the estimate of the proportion (50%) is quite different from the observed proportion (17%) and the confidence interval on the proportion is still within +/-5%

Overall, I think the sample size is adequate, but the assumptions of the sample size calculation are different from convention and from the observed proportion. A brief comment in the discussion on the adequacy of the sample size and differences between a priori assumed rate and observed rate may be considered.

Reply: we have included a brief comment on this under the first paragraph of the discussion part. Line # 322-323

The sampling strategy for the chart review is also clear and I congratulate the authors for describing this.

Reply: Thank you for your positive feedback 

Results.

All but two of the charts reviewed were of male patients?

This is surprising. Although there is a male predominance in VL, it is far from an exclusively male disease. (e.g., male/female ratio was 1.40 in PLOS NTD, Male predominance in reported Visceral Leishmaniasis cases: Nature or nurture? A comparison of population-based with health facility-reported data.)

The ratio in this study would be 296:2 !

I think this requires an explanation why female patients were not included (not likely chance alone).

Reply: This is an interesting point. Females were not excluded from the study. Significant male predominance of VL cases is the usual scenario, here in northwest Ethiopia. Our previous published reports also showed 97-100% of VL cases were male. Most of the VL cases in the northwest Ethiopia are seasonal migrant laborers who came from different parts of Ethiopia to harvest cash crops (and almost all are males). We discussed this briefly on the discussion part and put the references. Line # 373-376

Risk factors for AKI (Co-infection and HIV): please give odds ratio and 95%CI for the association between HIV and AKI, likewise for co-infections and AKI.

Reply: We run a multivariate regression analysis and we found that HIV status and other concomitant infections were significantly associated with the development of AKI. We provide the adjusted odds ratio and 95%. Line # 241-246. We have modified the result section of the abstract accordingly.

Editorial/language revisions:

clinician judgment of the treating physician: clinical judgment of the treating physician(Methods, Variables and operational definitions, paragraph 2)

Reply: corrected. Line # 154

Patient’s data confidentially were respected: patient data confidentiality was respected (Methods, ethical considerations)

Reply: sentence adjusted. Line # 178

“The main etiology of AKI at admission and during treatment course were dehydration and nephrotoxicity” – "nephrotoxicity" is not really an etiology (too general a term) – perhaps reword: medication-induced tubular injury or glomerular injury, for example. (Results)

Reply: we changed to” drug –induced AKI”- which is usually tubular injury

Table 2: AST elevation. LN (missing U for ULN) → corrected

---

## [Decision Letter · Decision Letter 1]

17 May 2021

Acute Kidney Injury in patients with Visceral Leishmaniasis in Northwest Ethiopia

PONE-D-21-00134R1

Dear Dr. Hailu,

We’re pleased to inform you that your manuscript has been judged scientifically suitable for publication and will be formally accepted for publication once it meets all outstanding technical requirements.

Kind regards,

Andrea L. Conroy, PhD

Academic Editor

PLOS ONE

Additional Editor Comments (optional):

Reviewers' comments:

Reviewer's Responses to Questions

**Comments to the Author**

1. If the authors have adequately addressed your comments raised in a previous round of review and you feel that this manuscript is now acceptable for publication, you may indicate that here to bypass the “Comments to the Author” section, enter your conflict of interest statement in the “Confidential to Editor” section, and submit your "Accept" recommendation.

Reviewer #1: All comments have been addressed

Reviewer #2: All comments have been addressed

2. Is the manuscript technically sound, and do the data support the conclusions?

Reviewer #1: Yes

Reviewer #2: Yes

3. Has the statistical analysis been performed appropriately and rigorously? 

Reviewer #1: Yes

Reviewer #2: Yes

4. Have the authors made all data underlying the findings in their manuscript fully available?

Reviewer #1: Yes

Reviewer #2: Yes

5. Is the manuscript presented in an intelligible fashion and written in standard English?

Reviewer #1: Yes

Reviewer #2: Yes

6. Review Comments to the Author

Reviewer #1: (No Response)

Reviewer #2: Thank you for thoroughly addressing my comments. All revisions have been done to my satisfaction. The manuscript is very interesting!

7. PLOS authors have the option to publish the peer review history of their article (what does this mean?). If published, this will include your full peer review and any attached files.

Reviewer #1: No

Reviewer #2: **Yes: **Michael Hawkes

---

## [Editor Report · Acceptance letter]

31 May 2021

PONE-D-21-00134R1 

Acute Kidney Injury in patients with Visceral Leishmaniasis in Northwest Ethiopia 

Dear Dr. Hailu:

I'm pleased to inform you that your manuscript has been deemed suitable for publication in PLOS ONE. Congratulations! Your manuscript is now with our production department. 

Kind regards, 

on behalf of

Dr. Andrea L. Conroy 

Academic Editor

PLOS ONE